# A Propensity Score-Matched Analysis to Assess the Outcomes in Pre- and Post-Fast-Track Hip and Knee Elective Prosthesis Patients

**DOI:** 10.3390/jcm10040741

**Published:** 2021-02-12

**Authors:** Luigi U. Romano, Marta Rigoni, Emanuele Torri, Marilena Nella, Monica Morandi, Piergiorgio Casetti, Giandomenico Nollo

**Affiliations:** 1Azienda Provinciale per i Servizi Sanitari, Ospedale di Tione, 38079 Trento, Italy; LuigiUmberto.Romano@apss.tn.it (L.U.R.); marilena.nella@apss.tn.it (M.N.); monica.morandi@apss.tn.it (M.M.); piergiorgio.casetti@apss.tn.it (P.C.); 2Fondazione Bruno Kessler, Healthcare Research and Implementation Program, 38123 Trento, Italy; giandomenico.nollo@unitn.it; 3Dipartimento di Ingegneria Industriale, Università degli studi di Trento, 38123 Trento, Italy; 4Dipartimento Salute e Politiche Sociali, Provincia autonoma of Trento, 38121 Trento, Italy; emanuele.torri@provincia.tn.it

**Keywords:** knee arthroplasty, hip arthroplasty, fast-track, enhanced recovery after surgery, propensity score, orthopedic care pathway

## Abstract

Fast-track surgery is a multimodal evidence-based approach to perioperative care aimed at reducing complications and recovery time. We compared a fast-track protocol to standard care in the setting of a small Italian general hospital. Propensity score estimation before and after the study was performed to compare pre-fast-track (pre-FT; January 2013–March 2014) and fast-track (FT; January 2016–December 2016) patients undergoing elective hip and knee replacement surgery with a three-year follow-up (up to January 2020). The primary endpoints were the mean hemoglobin drop, mean predischarge hemoglobin, transfusion and reinfusion rates, pain, ambulation day, hospital length of stay (LOS), and discharge to home/outpatient care or rehabilitation hospital center. The secondary endpoints were the adherence measures to the FT protocol, namely, tourniquet and surgical times, use of drains and catheters, type of anesthesia administered, and complications within three years. The risk difference (RD) and the adjusted odds ratio (aOR) were calculated for each outcome. After the propensity score estimation, we analyzed 59 patients in the pre-FT and 122 in the FT categories. The FT patients, with respect to the pre-FT patients, ameliorated their mean hemoglobin drop from 3.7 to 3.1 g/dl (*p* < 0.01) and improved their predischarge mean hemoglobin (10.5 g/dL versus 11.0 g/dL; *p* = 0.01). Furthermore, the aOR of being transfused was reduced by 81% (*p* < 0,01); the RD of being reinfused was reduced by 63% (*p* < 0.01); the aOR of having low pain on the first day was increased by more than six times (*p* < 0.01); the RD of ambulating the first day increased by 91% (*p* < 0.01); the aOR of admission to a rehabilitation hospital center was reduced by 98% (*p* < 0.01); the aOR of home discharge increased by 42 times (*p* < 0.01); the median LOS, tourniquet and surgical times, and use of catheters and drains significantly decreased. Patients with complications at 1 month were 43.1% and 38.2%, respectively, of pre-FT and FT patients (*p* = 0.63). Complications at 6, 12, 24, and 36 months were significantly lower for the FT patients. This study showed that the uptake of enhanced recovery practices was successful and resulted in the improvement of clinical and organizational outcomes. The fast-track concept and related programs may optimize perioperative care and streamline surgical and rehabilitation care paths.

## 1. Introduction

Hip and knee replacement operations are among the most commonly performed surgeries in developed countries [1,2]. Total hip arthroplasty (THA) and total knee arthroplasty (TKA), whose use is expected to increase further because of the aging population and the growing prevalence of osteoarthritis and other severe joint diseases, result in excellent functional improvement and pain relief outcomes. However, these procedures are subject to variation in quality and outcomes of care across providers and target populations. In addition, THA and TKA pose a serious risk of perioperative complications and are becoming costly due to the continuous advances in prosthetic design and materials [3,4,5].

Fast-track (FT) surgery, also known as enhanced recovery after surgery (ERAS), was pioneered in Denmark in the late 1990s by Professor Keleth for patients undergoing colorectal surgery [6] and was later successfully applied to a wide range of surgical disciplines, including orthopedics [7,8]. Fast-track surgery or ERAS may be defined as a coordinated perioperative approach aimed at reducing surgical stress and facilitating postoperative recovery [9]. This method is based on the implementation of a protocol encompassing several evidence-based elements, including patient education and engagement, mini-invasive surgery, multimodal anesthesia, pain control, fluid management, thromboembolic prophylaxis, appropriate wound management, blood conservation, early mobilization, and physiotherapy. In addition to these key elements, the care process should engage all of the professionals involved in the surgical and presurgical pathways, as well as the patients themselves [10].

Combining perioperative strategies for reducing surgical stress and enhanced postsurgical recovery in hip and knee arthroplasty has been shown to be associated with a shorter hospital stay, earlier achievement of functional milestones, and reduced complications, without an increase in readmission rates or compromising patient safety [11,12]. A recent paper even demonstrated that outpatient THA can be performed safely and successfully in a selected group of patients [13]. However, other authors stated that it is not yet clear whether outpatient TKA is worth considering, except in very exceptional cases (young patients without associated comorbidities). Outpatient TKA should not be generally recommended at the present time [14].

Therefore, the effectiveness of ERAS in arthroplasty has not been uniformly recognized or accepted by orthopedic surgeons [11]. The uptake of fast-track programs in clinical orthopedic practice has generally been slow, fragmented, and sparse [12,13,15,16]. Furthermore, evidence related to the introduction, execution, and sustained delivery of enhanced recovery programs and compliance with fast-track protocols is still challenging [17,18].

The purpose of our study was to evaluate short- and long-term outcomes, and compliance to surgical and non-surgical components after the implementation of a fast-track protocol for the elective treatment of hip and knee arthroplasty in the context of the care of a small public general hospital.

## 2. Methods

### 2.1. Design and Setting

We carried out a retrospective comparative observational study of patients admitted to the orthopedic and traumatology ward of Tione Hospital (Trento, Italy) undergoing an elective hip or knee arthroplasty replacement.

Tione Hospital is a first-level 60-bed hospital serviced by an emergency room, general medicine, general surgery, orthopedics and traumatology, a laboratory, diagnostic imaging, and rehabilitation services, and it serves a population of approximately 38,000 people in a mountainous region.

### 2.2. Implementation of the FT Protocol

The joint arthroplasty care team designed and implemented the FT protocol within the hospital. The joint arthroplasty care team was led by the chief orthopedic surgeon and included orthopedic surgeons, surgical assistants, anesthesiologists, nurse leaders, orthopedic nurses (and healthcare technicians), physical medicine and rehabilitation physicians, and physiotherapists.

The implementation of the FT bundle of practices was staggered. It started in March 2014 with the change of surgical techniques and bleeding management and gradually continued during 2015 with all other practices; special focus was on preoperative information and patients’ coaching, clinical assessment, anesthesia and pain control, wound management, and early mobilization. Therefore, the adoption of the new approach totaled 15–18 months, with the most active phase taking around 12 months. From January 2016, the hospital systematically applied the FT protocol, and all clinicians and nursing staff were engaged in the improvement process from the beginning. The FT protocol was set up and devised in accordance with the existing international recommendations and Italian practical guidance provided for hip and joint replacement surgery [19].

In the protocol, interventions were grouped according to different phases of the care path, as reported in Table 1, in which we compared the main pre-FT and FT practices.

It has been reported that a multimodal blood-loss prevention approach in primary TKA was highly effective, with a zero transfusion rate. Risk factors for transfusion were determined, and eliminating them contributed to the avoidance of allogeneic blood transfusion [20].

Throughout the implementation process, the promotion of teamwork and collaboration, training, and performance feedback were facilitated by external support for the assessment of the improvement processes and outcomes with the goal to define a standardized protocol for FT intervention for THA and TKA.

### 2.3. Study Population

All adult patients undergoing elective hip and knee arthroplasty admitted to the orthopedic and traumatology ward from 1 January 2013 to 31 March 2014 were included in the pre-FT period, and those admitted from 1 January 2016 to 31 December 2016 were treated as being in the FT period.

In the pre-FT population, we excluded patients treated for large prothesis revisions and unicompartmental knee arthroplasty (UKA). In the pre-FT period, ward organization and surgical and non-surgical care provision were conventional and did not incorporate any component linked to FT practice. During 2016, all patients followed the FT program. In order to analyze two homogeneous populations, between the two groups, we compared the following variables: age, sex, prothesis site, initial hemoglobin, and American Society of Anesthesiologists (ASA) score (according to the ASA physical status classification system).

We did not include patients admitted from March 2014 to December 2015, considering it a “switching period,” as the FT practices were under implementation and training.

### 2.4. Data Sources, Variables, and Outcomes

All demographic and clinical data were retrospectively collected. The sources of data and information were clinical records including all available information sources (i.e., medical and nursing assessments, administration records, operative checklists, discharge dispositions), and data from the hospital information system (collected at the time of the periodical examinations). Periodical examinations were performed after 1, 6, 12, 24, and 36 months following the intervention, for both the pre-FT and the FT patients.

We compared the pre-FT group versus the FT group on the following primary endpoints (clinical and organizational outcomes):mean hemoglobin drop;mean predischarge hemoglobin;number and percentages of hemotransfusions;number and percentages of blood reinfusions;first day pain score (number and percentage of patients with a numerical rating scale (NRS) < 4);accomplishment of postoperative ambulation;hospital length of stay;discharge setting (i.e., to home/outpatient rehabilitation or rehabilitation center).

We also selected a set of secondary endpoints as adherence measures to the FT protocol:median tourniquet time (knee replacement);median surgery time (hip and knee replacements);use of bladder catheter or surgical drain;type of anesthesia (i.e., subarachnoid, general, or combined anesthesia performed as selective subarachnoid anesthesia and short-acting sedative hypnotic agents, see Table 1).

The following long-term complications were evaluated at 1, 6, 12, 24, and 36 months after the intervention: pain, range of motion, infections, surgical wound, hypotonus, tendonitis, hematoma, joint effusion, periprosthetic joint infection, and thromboembolic events. Moreover, unplanned hospital readmissions after THA and TKA from all medical and surgical causes (including femur fracture, dislocation, prosthesis substitution, postoperative hematoma, surgical site infection, and dehiscence) within three years of discharge were identified from the administrative data and further evaluated by reviewing patients’ medical records.

According to the rules of the Healthcare Trust of the Autonomous Province of Trento, no ethical committee approval was needed since data were managed for clinical purposes by physicians and anonymized before statistical analyses in order to protect patient privacy.

### 2.5. Statistical Analyses

Categorical variables were expressed as frequencies and percentages, while continuous variables were expressed as means and standard deviations (SDs) or medians and first and third quartiles (Q1–Q3). We compared the differences in patient characteristics between the pre-FT and FT groups using Mann–Whitney *U* tests and Student’s *t*-tests, as well as Pearson’s chi-squared, Fisher’s exact, and chi-squared tests when appropriate.

To compare homogeneous groups of patients in terms of baseline covariates, a propensity score model was implemented [21]. Estimation of the propensity score was performed with a logistic regression model based on a region of common support, including age, site of replacement, presurgical hemoglobin, type of anesthesia, and ASA score as variables. Matching between the pre-FT and FT patients was done using the kernel matching method.

The process measures and outcomes were analyzed with a more appropriate statistical model or form of analysis, according to the nature of the indicator. Therefore, risk differences and crude and adjusted logistic regression models were performed. For each outcome, we calculated the effect measure for the FT period in comparison to the pre-FT period, adjusting for age, sex, and ASA score.

A *p*-value < 0.05 was considered statistically significant. We performed all analyses using Stata statistical software, version 13.0 (StataCorp, College Station, TX, USA).

## 3. Results

### 3.1. Patient Cohorts

The distribution of the baseline covariates before patient matching was not uniform for the ASA variable. A flow chart of the patient selection process is shown in Figure 1. The demographic and clinical characteristics of the pre-FT (*n* = 59) and FT (*n* = 122) patients after propensity score matching are presented in Table 2. As shown, the two groups were uniform in terms of the analyzed variables.

### 3.2. Short-Term Outcomes

Results on the clinical and organizational outcomes are shown in Table 3. All outcomes revealed a significant positive change moving from pre-FT to FT.

In relation to blood management practices, mean hemoglobin drop was reduced and predischarge hemoglobin was increased by the FT period. Treatment with FT reduced the odds of having a transfusion by 81% (aOR = 0.19, *p* < 0.01) with respect to the pre-FT patients. Similarly, the crude risk difference of being reinfused was reduced by 63% (RD = −0.63, *p* < 0.01) for the FT patients compared to pre-FT patients. After FT implementation, the probability of having lower pain (i.e., pain scale <4) on the first day increased by more than six times (aOR = 6.34, *p* < 0.01). In the FT population, the chance of ambulating within 24 h after the intervention increased by 91% (RD = 0.91, *p* < 0.01), and median LOS was reduced by the FT protocol. After FT implementation, the odds of discharge to a hospital rehabilitation service decreased by 98% (aOR = 0.02, *p* < 0.01), and the odds of home discharge increased by 42 times (aOR = 41.9, *p* < 0.01).

### 3.3. Compliance with the FT Protocol

We assessed the process measures as shown in Table 4. All measures demonstrated a significant change between the pre-FT and FT periods, confirming comprehensive compliance with the protocol and its more relevant practices. Both the median tourniquet and the surgery times decreased significantly in the FT period compared to the pre-FT period. Furthermore, both interquartile ranges decreased after the implementation of the FT protocol, indicating a more standardized procedure. The odds of receiving a bladder catheter were reduced by 81% for the FT patients compared to the pre-FT patients (adjusted odds ratio (aOR) = 0.19, *p* < 0.01), and the crude absolute risk of receiving a surgical drain was reduced by 92% (risk difference (RD) = −0.92, *p* < 0.01). In the FT patients, crude risk differences related to the given anesthesia showed reductions in subarachnoid and general anesthesia by 53% (RD = −0.53, *p* < 0.01) and 20% (RD = −0.20, *p* < 0.01), respectively; the use of combined anesthesia increased by 70% (RD = 0.70, *p* < 0.01).

### 3.4. Complications and Readmissions

The results of the long-term outcomes are presented in Table 5. According to chi-squared statistics, significant results in favor of FT were reported at 6, 12, 24, and 36 months. There were four hospital readmissions within 36 months among 59 patients in the pre-FT group, and there were four among 122 patients in the FT group (reduction was not significant). Causes of readmission for pre-FT were knee joint stiffness, hip joint replacement, hematoma, and periprosthetic joint infection; for FT, they were femur fracture, prosthesis mechanical complications, periprosthetic joint infection, and knee monoarthritis.

## 4. Discussion

In this retrospective study, we implemented a fast-track protocol intervention aimed at improving the care in routine elective total hip and knee replacement in the logistic and clinical setting of an orthopedic ward of a small general hospital. We demonstrated that the transition from pre-FT to FT was associated with meaningful improvement of clinical, functional, and organizational outcomes, and measurable compliance with enhanced recovery after surgery practices. We used a propensity score matching model to make the two patient populations comparable in terms of the baseline characteristics, as previously applied in similar studies [22,23]. The main findings of our study were an enhancement of intrahospital care and therapy, including improved pain control, better blood management, with a decrease in hemoglobin loss and a decreasing number of transfusions and blood reinfusions. Moreover, we demonstrated a reduction in hospital length of stay, and faster patient recovery and mobilization. Indeed, the results showed that remarkably fewer patients were admitted to rehabilitation hospital centers instead of home discharge and ambulatory physiotherapy.

Finally, in the mid-term and long term, we found a reduction in adverse outcomes and complications. Readmission rate was not affected by the FT program. Measures of adherence to the FT protocol, such as tourniquet and surgery times, use of bladder catheter and surgical drain, and type of anesthesia administered, decreased as expected by the effective implementation of the FT protocol.

Our findings are in agreement with several studies reinforcing evidence of the benefits for the patients and the healthcare services, although some inhomogeneity was present among studies on the FT protocol and outcome definitions, and on follow-up length. A number of papers indicated that the ERAS protocol reduced LOS without affecting major complications, hospital readmissions, and functional outcomes, as evaluated in a short-term (up to 3 months) [4,24] or long-term follow-up (up to 18 months) [25,26]. In addition, meta-analyses showed that ERAS significantly reduced the transfusion rate, incidence of complications, and LOS of patients undergoing TKA or THA, but it did not show a significant impact on functionality and 30-day readmission rate [27,28].

Our study also investigated the adherence to the FT protocol for perioperative steps, showing optimum compliance with FT elements and care phases. Indeed, the percentage of patients treated with selective subarachnoid anesthesia and short-acting sedative hypnotic agents increased in the FT group, leading to an improvement of patient comfort, a reduction of anxiety, and better controlled blood pressure levels with a positive benefit for bleeding. Furthermore, the use of short-acting hypnotic agents does not interfere with postoperative recovery, allowing early oral intake, early mobilization, and no bladder catheters. Moreover, as a consequence of adherence to a standardized process, surgical time and its variability among patients decreased significantly.

The entirety of our results demonstrated that a strict implementation of the FT protocol is also reliable in a first-level public hospital context, although it requires strong clinical leadership and engagement of all the medical and nursing staff to share a common philosophy, designing and educating people on the new practices. Indeed, the fast-track pathways rely on multidisciplinary teams and require coordinated interventions in all phases of perioperative care, from the initial preadmission consultation through to the hospitalization period and onward to the return of the patient to the normal activities of daily living [29]. In our experience, it took some time to reduce barriers to change and to motivate people. This caveat should be taken into account for the transferability of our results in different contexts.

## 5. Limitations

We are aware of the limitations of the retrospective nature and lack of control (both before and after the study) of our work. The clinical implications and contextual organizational constraints did not allow the development of a controlled random trial. It is known that when comparing different treatments, under real life conditions (i.e., outside the context of a randomized clinical trial), “confounding by indication” is a major problem that needs to be properly addressed [30]. In our study, we worked on an unselected patient population and we implemented a propensity score model to ensure matching of the two treatment groups in terms of demographics, age, and risk profiles. Where possible, adjusted logistic regressions were performed so as to consider of main risk factors.

The generalizability of the results of our study is limited by the sample size, duration, setting, target, and potential challenges in the execution of the FT protocol. On the other hand, the physical and organizational setting of care, the technologies, and the professionals themselves were unchanged from 2013 to 2016, and apart from the FT care path development and implementation, no other meaningful changes or actions occurred on the ward or in the hospital during the timeframe considered.

## 6. Conclusions

Our findings demonstrated the feasibility of a systematic assessment and evidence of positive outcomes. According to our results, the implementation of a bundle of evidence-based practices may enable the delivery of a superior care pathway for patients and healthcare services in short- and long-term periods. Considering the breadth and impact of hip and knee surgeries, even small improvements in perioperative care and the postoperative recovery process may have potential for delivering wide-ranging benefits.

## Figures and Tables

**Figure 1 jcm-10-00741-f001:**
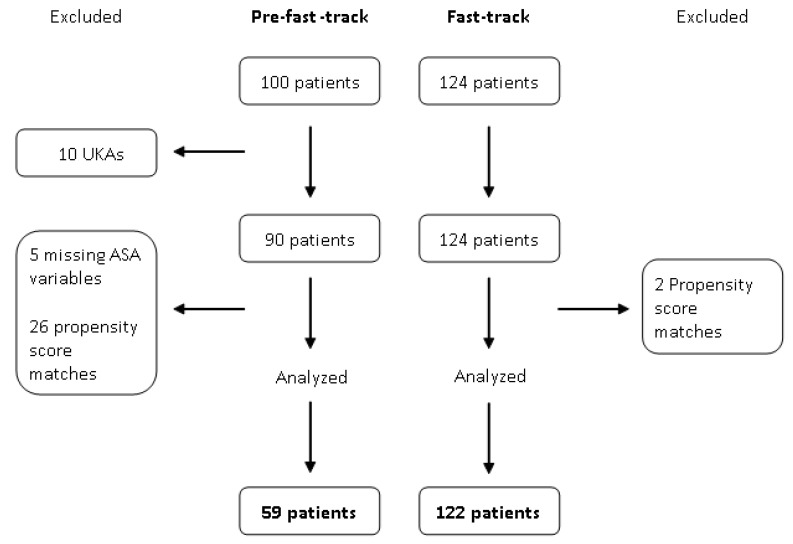
Flow chart reporting the patient selection process. The pre-FT and FT periods corresponded to 2013–2014 and 2016, respectively. ASA, American Society of Anesthesiologists; UKA, unicompartmental knee arthroplasty.

**Table 1 jcm-10-00741-t001:** Main characteristics of the fast-track (FT) protocol and the pre-fast-track (pre-FT) care at Tione Hospital, Trento (Italy).

FT Elements and Care Phases	FT Protocol Practices	Pre-FT
**Preadmission Care**		
Education and counseling	Patient engagement through multidisciplinary team, motivational coaching program, and psychological support Optimization of social back-up of the patient Optimization of fitness and lifestyle counseling Educational steps toward “functional milestones” with physiotherapist	Not present
Assessment and optimization of diseases/comorbidities	Preoperative health status assessment by multidisciplinary team (orthopedic surgeon, anesthesiologist, nurse, and physiotherapist) Risk categorization (i.e., infection, thromboembolic, drug tolerance, catheterization) Detection and treatment of chronic clinical conditions (e.g., diabetes and hypertension) and malnutrition Anemia workup and management	One week in advance: Chemical, biochemical, electrocardiogram, and imaging exams, as well as clinical consultation Assessment according to the ASA physical status classification system
**Preoperative and Intraoperative Care**		
Liberal fasting and carbohydrate loading	Minimal preoperative fasting	
Pre-emptive oral analgesia	Standard protocol (oxycodone naloxone)	No standard protocol
Standardized anesthetic protocol and multimodal analgesia	Selective subarachnoid anesthesia (i.e., Marcaine and levobupivacaine) Local infiltration analgesia (ropivacaine) Short-acting sedative hypnotic agents (i.e., midazolam and propofol)	Conventional anesthesia and sedation
Minimally invasive surgery	Standard technique with many evolutionary features; tissue-sparing approach Mini-posterolateral minimal incision and mini-medial parapatellar incision for knee Direct anterior approach for hip Barbed suture and Robert Jones bandage (knee) Surgical time management	Direct anterior or lateral approach according to the surgeon’s preferences Traditional suture
Blood conservation	Tranexamic acid (14 mg/kg pre-op, 14 mg/kg intra-op and 14 mg/kg post-op (regardless if mono- or bilateral); 3 g intra-articular for knee and 1.5 g intra-articular for hip; if bilateral contemporary surgery, half dose each side Reduced time and tourniquet compression (max 250 mmHg and max 40 min) Accurate hemostasis	Prestorage of three blood bags for autologous reinfusion Availability of three erythrocyte concentrates Non-systematic provision of hemostasis Drains and bladder catheters
Avoidance of drains and tubes	No drains or catheters (except for specific indications)	
Perioperative fluid management	Restricted fluid balance or zero balance regimen	1500 mL of crystalloid fluids on day 0
Prevention of hypothermia	Active patient and fluid warming Intraoperative warm blanket	If needed
**Postoperative care**		
Postoperative analgesia	Antalgic posture Multimodal opioid sparing analgesia (oxycodone–naloxone, NSAIDs, protonic pump inhibitors, paracetamol, intermittent cryocompression, etc.)	Peridural catheter Elastomer Opiates NSAIDs at fixed hours and therapy as needed (variable according to the prescriber)
Blood conservation and wound management	Intermittent cryocompression Daily wound management algorithm Negative pressure wound therapy if needed	Traditional dressing with medicated plaster; replaced every 2–3 days Suture with staples, removed after 10 days
Preventing and treating postoperative nausea and vomiting	Multimodal postoperative nausea and vomiting prophylaxis	Not present
Early oral intake	Regular diet within 4 h after surgery	24 h of fasting
Early mobilization andphysiotherapy	Day 0 (within 4 h): Milestones and verticalization and ambulation with walker (short journey) Day 1: Day 0 milestones and sitting in the chair and ambulation with walker (long journey) Day 2: Day 1 milestones and autonomous ambulation with crutches up and down the stairs Day 3: Day 2 milestones and stairs ambulation and consolidation of acquired competencies	Day 0: Bed rest Day 1: Assisted mobilization with a continuous passive motion device, assisted mobilization with physiotherapist, and mobilization from bed to chair Days 2–6: Ambulation using walker with partial weight-bearing load

Legend: ASA, American Society of Anesthesiologists; NSAIDs, nonsteroidal anti-inflammatory drugs.

**Table 2 jcm-10-00741-t002:** Baseline characteristics of the patients after the propensity score matching.

Characteristics	Pre-FT (59 Patients)	FT (122 Patients)	*p*-Value
Age, median (Q1–Q3), years	73 (68–77)	70 (64–77)	0.06
Male, *n* (%)	28 (47)	67 (55)	0.35
Female, *n* (%)	31 (53)	55 (45)	
Hip, *n* (%)	30 (51)	63 (52)	0.92
Knee, *n* (%)	29 (49)	59 (48)	
Presurgery Hb, mean (SD) g/dL	14.2 (1.3)	14.1 (1.3)	0.71
ASA physical status classification system 1, *n* (%)	11 (19%)	35 (29%)	0.15
ASA physical status classification system 2, *n* (%)	41 (69%)	66 (54%)	0.15
ASA physical status classification system 3, *n* (%)	7 (12%)	21 (17%)	0.15

Legend: FT, fast-track; Q1, first quartile; Q3, third quartile; Hb, hemoglobin; ASA, American Society of Anesthesiologists.

**Table 3 jcm-10-00741-t003:** Impact of the FT protocol on the outcomes of the two patient groups.

Outcomes	Pre-FT	FT	Effect Measures (95% CI)	*p*-Value
	59 patients	122 patients		
Mean Hb drop (g/dl), mean (SD)	3.7 (1.3)	3.1 (1.2)		<0.01 ^§^
Mean predischarge Hb (g/dl), mean (SD)	10.5 (1.1)	11.0 (1.3)		0.01 ^§^
Hemotransfusion, *n* (%)	26 (44)	17 (14)	0.19 (0.09–0.40) ^$^	<0.01
Blood reinfusion, *n* (%)	37 (63)	0 (0)	−0.63 (−0.75 to −0.50) ^*^	<0.01
Pain NRS scale <4 (first day), *n* (%)	21 (35)	94 (77)	6.34 (3.15–12.79) ^$^	<0.01
Patients ambulating in the first 24 h, *n* (%)	0 (0)	111 (91)	0.91 (0.86–0.96) ^*^	<0.01
Hospital length of stay, median (Q1–Q3)	8 (8–10)	5 (4–6)		<0.01 ^#^
Discharge to hospital rehabilitation center, *n* (%)	57 (97)	45 (37)	0.02 (0.01–0.06) ^$^	<0.01
Home discharge, *n* (%)	2 (3)	72 (59)	41.9 (12.1–144.9) ^$^	<0.01

Legend: aOR, adjusted odds ratio; CI, confidence interval; FT, fast-track; Hb, hemoglobin; LOS, length of stay; NRS, numerical rating scale (0 = no pain, 10 = worst pain); Q1, first quartile; Q3, third quartile; ^§^ Student’s *t*-test; ^#^ Mann–Whitney *U* test; ^$^ adjusted odds ratio; * crude risk difference.

**Table 4 jcm-10-00741-t004:** Impact of the FT protocol on the process of care of the two patient groups.

Measure	Pre-FT	FT	Effect Measures (95% CI)	*p*-Value
	59 patients	122 patients		
Tourniquet time, median (Q1–Q3), minutes	61 (54–69.5)	35 (30–41)		<0.01 ^#^
Surgery time, median (Q1–Q3), minutes	106 (94–115)	86 (77–92)		<0.01 ^#^
Bladder catheter, *n* (%)	16 (27)	10 (8)	0.19 (0.08–0.43) ^$^	<0.01
Surgical drain, *n* (%)	54 (92)	0 (0)	−0.92 (−0.99 to −0.84) *	<0.01
Subarachnoidanesthesia, *n* (%)	34 (57)	6 (5)	−0.53 (−0.66 to −0.40) *	<0.01
General anesthesia, *n* (%)	17 (28)	11 (10)	−0.20 (−0.32 to −0.07) *	<0.01
Combined anesthesia, *n* (%)	9 (15)	104 (85)	0.70 (0.59 to 0.81) *	<0.01

Legend: CI, confidence interval; FT, fast-track; Q1, first quartile; Q3, third quartile; ^#^ Mann–Whitney *U* test; ^$^ adjusted odds ratio; * crude risk difference. Combined anesthesia was defined as selective subarachnoid anesthesia and short-acting sedative hypnotic agents.

**Table 5 jcm-10-00741-t005:** Impact of the FT protocol on long-term complications and readmissions of the two patient groups.

Outcomes	Pre-FT	FT	Crude OR (95% CI)	*p*-Value
Patients with 1 or more complications	59 patients	122 patients		
1 month, *n* (%)	25 (43.1)	47 (38.2)	0.85 (0.45–1.61) OR	0.63
6 months, *n* (%)	14 (23.7)	10 (8.0)	0.29 (0.12–0.67) OR	<0.01
12 months, *n* (%)	15 (25.4)	6 (4.9)	0.15 (0.06–0.38) OR	<0.01
24 months, *n* (%)	11 (18.6)	2 (1.6)	0.07 (0.02–0.25) OR	<0.01
36 months, *n* (%)	9 (15.2)	1 (0.8)	0.05 (0.01–0.21) OR	<0.01
Hospital readmissions within 36 months, *n* (%)	4 (6.8)	4 (3.3)	0.46 (0.11–1.88) OR	0.44

Legend: CI, confidence interval; FT, fast-track; OR, odds ratio. Complications were computed as pain, range of motion, infections, surgical wound, hypotonus, tendonitis, hematoma, joint effusion, periprosthetic joint infection, and thromboembolic events.

## Data Availability

The data presented in this study are available on request from the corresponding author. The data are not publicly available due to privacy restrictions.

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
