# Peer review of "A Propensity Score-Matched Analysis to Assess the Outcomes in Pre- and Post-Fast-Track Hip and Knee Elective Prosthesis Patients"

_jcm, 2021, doi:10.3390/jcm10040741_

Round 1
Reviewer 1 Report
This interesting study includes most important issues related to fast-track total joint replacemenent. Particularly, the authors report the evolution of this perioperative management with a mid-term follow-up. So, despite Introduction and Methods are very well described, the reader could miss long-term outcome, please read below:
- Lines 218 to 224. There is no relevant information for complications between groups (infection, periprosthetic fracture, loosening, ...). I suggest to include (maybe also a Table) before this study is suitable for the Journal. Moreover, consider to comment this issue in Discussion
Author Response
We thank the reviewer for the careful reading of the manuscript and constructive remarks.
In the revised manuscript, we added a new Table (Table 5) to facilitate the reading of long term outcomes (complications and readmissions). Relevant information for complications was described in the “Data sources, variables, and outcomes” paragraph, further we added the information in the Legend of Table 5. Moreover, we addressed this issue in the Discussion also reporting short- and long-term results from literature.
Reviewer 2 Report
The authors describe the implementation of a fast track program in their hospital. Good results are presented. However, fast tract surgery has been well studied in the literature and the benefits and disadvantages are well understood. No advancement in current knowledge is made with this manuscript.
The introduction is rather big. Please make it shorter. The methods are well presented. Results are clearly presented and tables are ok.
The discussion needs to be rewritten. The first paragraph of the discussion correctly presents the results of the study. Please discuss the significance of the results. Suggestion to discuss the current literature and how contrasts/compares with the study. Discuss also the external validity of the study.
abstract, tables, keywords ok
specific points:
lines 45-47: The sentence makes no sense. Please rephrase. The corresponding references may not be relevant.
line 62: please explain "stress-reduced" surgery
line 134: please specific state the primary and secondary outcomes as in abstract
line 146: instead of thrombophlebitis use the word thromboembolic events
line 154: statistics: please state how the sample size was arrived. Was the sample powered enough? If the primary outcome was hemoglobin drop, difference 0.5 (SD =1) may be considered clinical significant and the power of the study can be calculated accordingly (Martin J.G., Cassatt K.B., Kincaid-Cinnamon K.A., Westendorf D.S., Garton A.S., Lemke J.H. Topical Administration of Tranexamic Acid in Primary Total Hip and Total Knee Arthroplasty. J. Arthroplast. 2014;29:889–894. doi: 10.1016/j.arth.2013.10.005)
line 195: the combined anaesthesia was increased. Please discuss the benefit of combined anaesthesia in the discussion
line 218: suggestion to write complications and readmissions instead of long-term outcomes as the title of the paragraph
A table with all complications and readmissions and with percentages in each group would improve the manuscript. P values can be presented as well
line 261: Postoperative complications are not adequacy described. As mentioned in the results section a table would be more suitable
line 279: The aim of the study should be stated in the introduction
Author Response
The authors describe the implementation of a fast track program in their hospital. Good results are presented. However, fast tract surgery has been well studied in the literature and the benefits and disadvantages are well understood. No advancement in current knowledge is made with this manuscript.
Answer:
We thank the reviewer for the careful reading of the manuscript and constructive remarks. Although fast track surgery has been well studied in the literature, we think that our paper could bring useful evidence. In fact, the strengths of this work for which we believe it deserves to be published are: firstly, the setting of a small rural public hospital (FT applications in such context are still rare and unpublished for most countries), secondly we assessed a set of outcomes covering all aspects of the fast track process of care, thirdly we measured the adherence to the applied protocol deriving some improvement lessons. Moreover, although the analysis is a retrospective one, the propensity score matching population increased the robustness of the results, and the physical and organizational setting of care, the technologies, and the professionals themselves were unchanged from 2013 to 2016, and apart from the FT care path development and implementation, no other meaningful changes or actions occurred on the ward or in the hospital during the timeframe considered. The latter aspect further strengthened the results.
The introduction is rather big. Please make it shorter. The methods are well presented. Results are clearly presented and tables are ok.
Answer:
As suggested, we shortened the introduction.
The discussion needs to be rewritten. The first paragraph of the discussion correctly presents the results of the study. Please discuss the significance of the results. Suggestion to discuss the current literature and how contrasts/compares with the study. Discuss also the external validity of the study.
Answer:
The Discussion was deeply revised focusing on results and comparison with current literature. The transferability of the results and study limitation were addressed in the Discussion and Limitations sections.
abstract, tables, keywords ok
specific points:
lines 45-47: The sentence makes no sense. Please rephrase. The corresponding references may not be relevant.
Answer:
We rephrased the sentence. References were refreshed along all the text.
line 62: please explain "stress-reduced" surgery
Answer:
We rephrased and clarified the sentence.
line 134: please specific state the primary and secondary outcomes as in abstract
Answer:
Following the suggestion, we used same definitions as stated in the abstract.
line 146: instead of thrombophlebitis use the word thromboembolic events
Answer:
In the revised manuscript, we used the word “thromboembolic events”.
line 154: statistics: please state how the sample size was arrived. Was the sample powered enough? If the primary outcome was hemoglobin drop, difference 0.5 (SD =1) may be considered clinical significant and the power of the study can be calculated accordingly (Martin J.G., Cassatt K.B., Kincaid-Cinnamon K.A., Westendorf D.S., Garton A.S., Lemke J.H. Topical Administration of Tranexamic Acid in Primary Total Hip and Total Knee Arthroplasty. J. Arthroplast. 2014;29:889–894. doi: 10.1016/j.arth.2013.10.005)
Answer:
The aim of our study was to assess the impact of the implementation of a fast-track protocol for THA and TKA in elective patients. Figure 1 sowed the flow chart of patient selection process. We decided to include all patients treated in the Tione public hospital in 2013-2014 and 2016. Some exclusions were made because of unicompartmental knee arthroplasty, 5 patients with missing data, and propensity score matching between groups. We opted to evaluate a set of clinical and organizational outcomes and a set of process outcomes (i.e. adherence to the FT protocol). In other words, the study did not have a single primary outcome, but a set of primary outcomes. Moreover, from the methodological point of view, the study was a retrospective and observational one and the study was not designed to test a null hypothesis on a single outcome (for example Administration of Tranexamic Acid to reduce bleeding). According to these considerations, a sample size calculation/power calculation based on a sole outcome is not appropriate due to the nature and aim of our study.
line 195: the combined anaesthesia was increased. Please discuss the benefit of combined anaesthesia in the discussion
Answer:
We detailed benefits of combiner anesthesia (i.e. selective subarachnoid anesthesia and short-acting sedative hypnotic agents) in the Discussion.
line 218: suggestion to write complications and readmissions instead of long-term outcomes as the title of the paragraph
Answer:
According to Reviewer’s suggestion, we changed the title of the paragraph.
A table with all complications and readmissions and with percentages in each group would improve the manuscript. P values can be presented as well
Answer:
In the revised manuscript, we added a new Table (Table 5) with numbers, percentages and p-value for complications and readmissions. Thanks to this suggestion, reading and understanding the results is now easier.
line 261: Postoperative complications are not adequacy described. As mentioned in the results section a table would be more suitable
Answer:
In the revised manuscript, we added a new Table (Table 5) with numbers, percentages and p-value for complications and readmissions. Relevant information for complications were described in the “Data sources, variables, and outcomes” paragraph, further we added the information in the Legend of Table 5.
line 279: The aim of the study should be stated in the introduction
Answer:
We cancelled the sentence.
Reviewer 3 Report
Interesting paper on the benefits of fast track in TKA and THA. However, the following minor changes are required:
+In line 218 (section 3.4 Long-term outcomes) the authors mention many complications: 39/59 (66%) in the Pre-FT group and 78/122 (63%) in the FT group. Was that difference statistically significant as the authors mention in their paper? Please, check. The authors must explain in detail what these complications were and how they solved them, and also why they think they have such a high rate of complications.
+ In the ABSTRACT the high rate of complications must be also mentioned.
+ In line 52, together with reference [8] the authors should include a new one. It is the following:
Rodríguez-Merchán EC. Pros and cons of fast-track total knee arthroplasty. Int Journal of Orthopaedics 2015 June 23; 2(3):270-279. ISSN 2311-5106 (Print), ISSN 2313-1462 (Online).
+ In line 66, after reference [14] the following sentence (and reference) must be included: “However, other paper has stated that it is not yet clear whether outpatient TKA is worth considering, except in very exceptional cases (young patients without associated comorbidities). Outpatient TKA should not be generally recommended at the present time”. [Rodríguez-Merchán EC. Outpatient total knee arthroplasty: is it worth considering? EFORT Open Rev. 2020 Mar 2;5(3):172-179. doi: 10.1302/2058-5241.5.180101. eCollection 2020 Mar.PMID: 32296551].
+ In line 102, at the end of the sentence a new sentence (and reference) must be included as follows: “It has been reported that a multimodal blood-loss prevention approach in primary TKA was highly effective, with a zero transfusion rate. Risk factors for transfusion were determined, and eliminating them contributed to the avoidance of allogeneic blood transfusion [Moráis S, Ortega-Andreu M, Rodríguez-Merchán EC, Padilla-Eguiluz NG, Pérez-Chrzanowska H, Figueredo-Zalve R, Gómez-Barrena E. Blood transfusion after primary total knee arthroplasty can be significantly minimised through a multimodal blood-loss prevention approach. Int Orthop. 2014 Feb;38(2):347-54. doi: 10.1007/s00264-013-2188-7. Epub 2013 Dec 7.PMID: 24318318].
Author Response
Interesting paper on the benefits of fast track in TKA and THA. However, the following minor changes are required:
+In line 218 (section 3.4 Long-term outcomes) the authors mention many complications: 39/59 (66%) in the Pre-FT group and 78/122 (63%) in the FT group. Was that difference statistically significant as the authors mention in their paper? Please, check. The authors must explain in detail what these complications were and how they solved them, and also why they think they have such a high rate of complications.
Answer:
We thank the reviewer for the careful reading of the manuscript and constructive remarks.
As far as the complications are concerning, in the previous version of the manuscript we reported the total number of complications among patients, indicating 66.1% and 63.9%, p=0.46, respectively for pre-FT and FT at 1 month. We thought that calculation might be misleading, thus, in the revised paper we have reported more simply the number of patients with 1 or more complications. As detailed in the text, complications were pain, range of motion, infections, surgical wound, hypotonus, and joint effusion for each patient. The percentages of patients with 1 or more complications at 1 month were 43.1% and 38.2% (p=0.63) respectively for pre-FT and FT patients.
As reported, these complications were almost solved at 6 months. The comparison of our results with current literature is not trivial as “complications” are defined in very different ways among studies and many of them are qualitative evaluations. However, we want to underline that the paper was focused on the outcome comparison between pre-FT and FT groups. Our results demonstrated improved outcomes for FT patients in respect to pre-FT, measuring outcomes in the same way, by the same staff.
To facilitate the reading and comprehension of complications, we added a Table (Table 5) with numbers, percentages and p-values. Relevant information for complications were previously described in the “Data sources, variables, and outcomes” paragraph, further we added the information also in the Legend of Table 5.
+ In the ABSTRACT the high rate of complications must be also mentioned.
Answer:
We added this outcome information in the abstract.
+ In line 52, together with reference [8] the authors should include a new one. It is the following:
Rodríguez-Merchán EC. Pros and cons of fast-track total knee arthroplasty. Int Journal of Orthopaedics 2015 June 23; 2(3):270-279. ISSN 2311-5106 (Print), ISSN 2313-1462 (Online).
Answer:
The reference was added.
+ In line 66, after reference [14] the following sentence (and reference) must be included: “However, other paper has stated that it is not yet clear whether outpatient TKA is worth considering, except in very exceptional cases (young patients without associated comorbidities). Outpatient TKA should not be generally recommended at the present time”. [Rodríguez-Merchán EC. Outpatient total knee arthroplasty: is it worth considering? EFORT Open Rev. 2020 Mar 2;5(3):172-179. doi: 10.1302/2058-5241.5.180101. eCollection 2020 Mar.PMID: 32296551].
Answer:
Sentence and reference were added.
+ In line 102, at the end of the sentence a new sentence (and reference) must be included as follows: “It has been reported that a multimodal blood-loss prevention approach in primary TKA was highly effective, with a zero transfusion rate. Risk factors for transfusion were determined,and eliminating them contributed to the avoidance of allogeneic blood transfusion [Moráis S, Ortega-Andreu M, Rodríguez-Merchán EC, Padilla-Eguiluz NG, Pérez-Chrzanowska H, Figueredo-Zalve R, Gómez-Barrena E. Blood transfusion after primary total knee arthroplasty can be significantly minimised through a multimodal blood-loss prevention approach. Int Orthop. 2014 Feb;38(2):347-54. doi: 10.1007/s00264-013-2188-7. Epub 2013 Dec 7.PMID: 24318318].
Answer:
Sentence and reference were added.
Round 2
Reviewer 2 Report
Improved manuscript after revision.
Minor points:
line 142: please explain how the surgical time can be a utilised as endpoint for measuring adherence to FT protocol? How was surgical time management implemented?
lines 195-196: "After FT implementation, the odds of having a pain scale <4 was reduced by 87% (aOR = 0.13, p < 0.01)"
This is in contrast with table 3. Please rephrase
Author Response
Reviewer 2
Improved manuscript after revision.
Answer:
We thank the Reviewer for the evaluation of the manuscript and its review.
Minor points:
line 142: please explain how the surgical time can be a utilised as endpoint for measuring adherence to FT protocol? How was surgical time management implemented?
Answer:
The surgical time can be considered as an endpoint of FT approach as it represents the result of all the coordinated actions taken upstream of the surgical act, and during the intervention (Preoperative & intraoperative care elements of the FT protocol, Table 1). The adherence of all the protocol steps according to a pre-settled standard is expected to reduce surgical time and surgical time variability among patients. From the clinical point of view, the containment of surgical time contributes to the reduction of complications, peri-operative risks, and to the early recovery. Hence, the surgical time can be considered as a proxy measure of efficiency (e.g. standardization of the processes). As reported in the original manuscript, the reduction of surgery time was accompanied with the reduction of the interquartile range, indicated a more standardized procedure (lines 211-212).
These concepts, in the revised version of the manuscript, were briefly addressed in the Methods (lines 106-107) and Discussion (lines 267-268) sections.
lines 195-196: "After FT implementation, the odds of having a pain scale <4 was reduced by 87% (aOR = 0.13, p < 0.01)"
This is in contrast with table 3. Please rephrase
Answer:
We thank the reviewer for pointing out the error. We fixed the aOR in Table 3, which is now coherent with the outcome definition (aOR= 6.34 (3.15–12.79) p<0.01). The sentence was rephrased accordingly with results in Table 3: “After FT implementation, the probability of having lower pain (i.e. pain scale <4) at the first day increased by almost six times (aOR = 6.34, p < 0.01).”, lines 196-197.